# Matrix Adaptation Evolution Strategy with Multi-Objective Optimization for Multimodal Optimization

**Wei Li** [1,2] 

[1] School of Computer Science and Engineering, Xi'an University of Technology, Xi'an 710048, China; liwei@xaut.edu.cn

[2] Shaanxi Key Laboratory for Network Computing and Security Technology, Xi'an 710048, China

**Abstract:** The standard covariance matrix adaptation evolution strategy (CMA-ES) is highly effective at locating a single global optimum. However, it shows unsatisfactory performance for solving multimodal optimization problems (MMOPs). In this paper, an improved algorithm based on the MA-ES, which is called the matrix adaptation evolution strategy with multi-objective optimization algorithm, is proposed to solve multimodal optimization problems (MA-ESN-MO). Taking advantage of the multi-objective optimization in maintaining population diversity, MA-ESN-MO transforms an MMOP into a bi-objective optimization problem. The archive is employed to save better solutions for improving the convergence of the algorithm. Moreover, the peaks found by the algorithm can be maintained until the end of the run. Multiple subpopulations are used to explore and exploit in parallel to find multiple optimal solutions for the given problem. Experimental results on CEC2013 test problems show that the covariance matrix adaptation with Niching and the multi-objective optimization algorithm (CMA-NMO), CMA Niching with the Mahalanobis Metric and the multi-objective optimization algorithm (CMA-NMM-MO), and matrix adaptation evolution strategy Niching with the multi-objective optimization algorithm (MA-ESN-MO) have overall better performance compared with the covariance matrix adaptation evolution strategy (CMA-ES), matrix adaptation evolution strategy (MA-ES), CMA Niching (CMA-N), CMA-ES Niching with Mahalanobis Metric (CMA-NMM), and MA-ES-Niching (MA-ESN).

**Keywords:** multimodal optimization problems; multi-objective optimization; matrix adaptation evolution strategy; non-dominated sorting

---

## 1. Introduction

Many problems from the real world are classified as optimization problems. Some optimization problems that have one global solution are called single modal optimization problems, while others that have multiple global and local optima are known as multimodal optimization problems. Traditional evolutionary algorithms (EAs) are effective at converging to a single global optimum because of the global selection strategy used. However, it is inappropriate for EAs to solve multimodal optimization problems. In order to overcome the weakness, niching techniques are incorporated into EAs, such as differential evolution [1,2], particle swarm optimization [3], the covariance matrix adaptation evolution strategy (CMA-ES) [4], self-adaptive niching CMA-ES [4], and genetic algorithm [5], to solve multimodal optimization problems. The representative niching strategies include crowing [6], restricted tournament selection [7], fitness sharing [8], clearing [9], and speciation [5].

The covariance matrix adaptation evolution strategy (CMA-ES), which was proposed by Hansen and Ostermeier [10], is one of the popular optimization algorithms for solving unconstrained

real-parameter optimization. Different from other optimization algorithms, CMA-ES makes use of two evolution paths to realize exploitation and exploration during the search process. The two evolution paths are the learning of the mutation strength and the rank-1 update of the covariance matrix, respectively. The self-adaptively updated covariance matrix, which uses evolution path information, can be considered a time series prediction of the evolution of the parent [11]. Since CMA-ES employs the covariance matrix to exploit the information from the previous and current generations, it has attracted broad investigation in recent years. However, compared with other classical algorithms, such as DE or PSO, CMA-ES is slightly more complicated, because it has two evolution paths and an update of the covariance matrix. To simplify the standard CMA-ES, Beyer and Sendhoff proposed the matrix adaptation evolution strategy (MA-ES) [11], in which one of the evolution paths (namely the p-evolution path) is dropped, and the covariance matrix (namely the **C** matrix) is discarded. The experimental results in [11] show that the MA-ES exhibits a similar performance as the CMA-ES, which considers both standard population sizes ($\lambda < N$) and large population sizes ($\lambda = O(N^2)$).

The MA-ES only simplifies the CMA-ES. The performance of the MA-ES is basically the same as that of the CMA-ES. Therefore, the MA-ES is a robust local search strategy that efficiently solves unimodal optimization problems. It is unable to find multiple solutions in multimodal problems because of the designed parameters and updating rules [12]. At present, to the best of our knowledge, no work has been reported on utilizing the MA-ES to solve the multimodal problems. There has been an effort to provide two versions of CMA-ES, which are called the niching covariance matrix adaptation evolution strategy and the self-adaptive niching CMA-ES, respectively, for solving multimodal problems [13,14]. The two improved versions of the CMA-ES introduced niching strategies that can maintain the population diversity and realize parallel convergence within some subpopulations to obtain multiple good solutions. However, the performance of the two improved versions of CMA-ES is highly sensitive to niching parameters. So far, some works have been done to convert a multimodal optimization problem (MMOP) to a multi-objective optimization problem (MOP) [15–19]. The advantage of transforming an MMOP into an MOP is that it is unnecessary to use the problem-dependent niching parameters. However, the prerequisite for MOP is objective confliction, which makes it difficult to transform an MMOP into a multi-objective optimization problem [17]. To address this issue, this paper proposes an improved algorithm based on the MA-ES called the matrix adaptation evolution strategy with multi-objective optimization algorithm (MA-ESN-MO). The main contributions of this paper are summarized as follows:

1.  The MMOP is transferred into two objective optimization problems with strong objective confliction; the advantage of multi-objective optimization can be fully used to ensure the diversity of the population.
2.  The information of the population landscape and the fitness of the objective function are employed to construct two conflicting objective functions instead of utilizing classical niching strategies. Moreover, the archive is employed to save better individuals, which are helpful to ensure the convergence of the algorithm. In this manner, the exploration and exploitation abilities of the algorithm are balanced effectively.
3.  The population is divided into several subpopulations in MA-ES instead of using one population in CMA-ES. In this way, the algorithm can explore and exploit in parallel within these subpopulations to find multiple optimal solutions for the given problem. Moreover, the niching method is employed to improve the diversity of the population.
4.  Systematic experiments conducted to compare the algorithms including CMA-ES, MA-ES, CMA-ES-Niching (CMA-N), CMA-ES-Niching-MO (CMA-NMO), CMA-ES Niching with Mahalanobis Metric [20] (CMA-NMM), CMA-NMM-MO, MA-ES-Niching (MA-ESN), and MA-ESN-MO on CEC2013 multimodal benchmark problems [21] are described. CMA-NMO and CMA-NMM-MO are obtained by introducing the proposed method into CMA-ES and CMA-NMM. The experimental results show that the proposed method is promising for solving multimodal optimization problems.

The rest of this paper is organized as follows. In Section 2, the related work on multimodal optimization problems is reviewed. Section 3 introduces variants of CMA-ES and the framework of MA-ES. The proposed matrix adaptation evolution strategy with the multi-objective optimization algorithm (MA-ESN-MO) is presented in Section 4. Section 5 reports and discusses the experimental results. Finally, the conclusions and possible future research are drawn up in Section 6.

## 2. Related Work

Many evolutionary algorithms (EAs) can effectively solve single-objective optimization problems that involve only one optimal solution. However, they are unable to perform well on multimodal optimization problems because of their poor population diversity preservation. To address the issue, sufficient works have been done over the past decades. The strategies on improving the EAs fall into three categories [22].

Niching is an effective method that is used to find and preserve multiple stable niches for multimodal optimization problems. Classical niching techniques include crowding, fitness sharing, speciation, clearing, and restricted tournament selection. The two classic crowding strategies are deterministic crowding and probabilistic crowding. The deterministic crowding can effectively solve the problem of the replacement error, which is the main disadvantage of crowding, while probabilistic crowding utilizes the probabilistic selection to prevent the loss of niches with lower fitness or the loss of local optima [23–25]. Both the fitness sharing strategy and speciation divide the population into several subpopulations according to the similarity of the individuals, which can form and maintain the stable niches [5,8]. However, the niche radius $\sigma_{share}$ and $r_s$ that are used in sharing and speciation, respectively, are difficult to define because they require prior knowledge of the problems. The clearing strategy preserves the best individuals and removes the bad individuals of the niches during the generations [26]. However, the individuals will move toward the best individual area, which may arouse stagnates in the niching because of diversity loss. Restricted tournament selection [7] utilizes Euclidean or Hamming distance to find the nearest member within the $w$ (window size) individuals. The nearest member will compete with the offspring, and the winner will survive in the next generation.

The second strategy aims to enhance population diversity by introducing novel operators into EAs. Among some representative works, Hui et al. [27] proposed an ensemble and arithmetic recombination-based speciation DE (EARSDE) algorithm, where the arithmetic recombination with speciation is used to enhance exploration, and the neighborhood mutation with ensemble strategies is employed to improve the exploitation of individual peaks. Yang et al. [28] proposed an adaptive multimodal continuous ant colony optimization algorithm, where a local search scheme based on Gaussian distribution is used to enhance the exploitation and a differential evolution mutation operator is employed to accelerate convergence. Haghbayan et al. [29] proposed a niche gravitational search algorithm (GSA) method, where a nearest neighbor scheme and the hill valley algorithm are used to enable the species to explore more optima via diversity conservation in the swarm. Qu et al. [30] proposed a distance-based locally informed particle swarm optimization (PSO) algorithm, where several local best particles are used to guide the search of each particle instead of using the global best particle.

The third strategy is to introduce a novel transformation mechanism, in other words, a multimodal optimization problem (MMOP) is transformed into a multi-objective optimization problem (MOP) [17]. Among some representative works, Cheng et al. [22] proposed an evolutionary multi-objective optimization-based multimodal optimization algorithm, where approximate multimodal fitness landscapes are used to provide an estimation of potential optimal areas. Moreover, an adaptive peak detection strategy is employed to find peaks where optimal solutions may exist. Yu et al. [19] proposed a tri-objective differential evolution approach algorithm, where three optimization objectives are constructed to ensure good population diversity. In addition, a solution comparison rule and a ranking strategy are employed to enhance the accuracy of solutions. Wang et al. [17] proposed a multi-objective optimization for multimodal optimization problems (MOMMOP) algorithm, where an MMOP is transformed into a multi-objective optimization problem with two conflicting

objectives. Basak et al. [15] proposed a novel multimodal optimization algorithm, where a novel bi-objective formulation of the multimodal optimization problem and differential evolution (DE) with a non-dominated sorting strategy are used to detect multiple global and local optima. Deb et al. [16] proposed a bi-objective evolutionary algorithm, where the single-objective multimodal optimization problem is converted into a suitable bi-objective optimization problem to find multiple peaks. Yao et al. [18] proposed a multipopulation genetic algorithm, where a multipopulation and clustering scheme are used to improve exploitation within promising areas.

## 3. Variants of CMA-ES and MA-ES Algorithm

This section briefly reviews the improvements on CMA-ES and the framework of the MA-ES algorithm, respectively.

### 3.1. Variants of CMA-ES

In recent years, many CMA-ES variants have been developed to solve single-objective and multi-objective optimization problems. For single-objective optimization problems, a DE variant with covariance matrix self-adaptation (DECMSA) [31] combines different features of DE and CMA-ES instead of hybridizing these two algorithms simply. In DECMSA, the individuals are sampled from a Gaussian distribution to guide the search direction in the DE. In addition, an enhanced local search strategy based on CMA-ES is designed. In order to improve the classical optimization algorithm's ability of rotational invariance, differential evolution based on covariance matrix learning and the bimodal distribution parameter setting (CoBiDE) [32] utilizes the covariance matrix learning to construct an appropriate coordinate system for the crossover operator, while biogeography-based optimization (BBO) with covariance matrix-based migration (CMM-BBO) [33] utilizes the covariance matrix migration to reduce BBO's dependence on the coordinate system. The differential covariance matrix adaptation evolutionary algorithm (DCMA-EA) [34] incorporated the mutation, crossover, and selection operators of the DE into CMA-ES to compose a hybrid algorithm in order to improve the performance of CMA-ES for solving the optimization problems with complicated fitness landscapes. The bilevel covariance matrix adaptation evolution strategy (BL-CMA-ES) [35] employs CMA-ES at the upper level and lower level to extract the priori knowledge: the search distribution, which significantly reduces the number of function evaluations and improves the efficiency of the algorithm. The Differential Crossover Strategy based on Covariance Matrix Learning with Euclidean Neighborhood (L-covnSHADE) [36] employs covariance matrix learning to establish a coordinate system for the better crossover operator.

For multi-objective optimization problems (MOPs), multi-objective differential evolution (MODE) with dynamic covariance matrix learning (MODEs + DCML) [37] utilizes the dynamic covariance matrix learning (DCML) to establish a proper coordinate system for the binomial crossover operator. Therefore, the poor performance of MOPs solved by MODE can be significantly improved. Decomposition-based multi-objective evolutionary algorithms (MOEA/D)-CMA-ES [38] integrates CMA-ES into the decomposition-based multi-objective evolutionary algorithms (MOEA/D) to solve the multi-objective optimization problems. The hybrid algorithm MOEA/D-CMA-ES takes advantage of the MOEA/D in multi-objective optimization and CMA-ES in complex numerical optimization. The experimental results show that MOEA/D–CMAES is an effective algorithm for solving complex multi-objective optimization problems.

### 3.2. MA-ES Algorithm

The CMA-ES employs two evolution paths: one for the update of the covariance matrix and another for the learning of the covariance matrix. This increases the complexity of the algorithm. To solve this problem, the MA-ES drops one of the evolution paths and removes the covariance matrix. Briefly speaking, the MA-ES directly learns the **M** matrix from the path cumulation information.

Different from CMA-ES, the offspring of the new generation $g + 1$ is calculated by weighted recombination, which is expressed by the following formula:

$$\mathbf{y}^{g+1} = \mathbf{x}^g + \sigma^g \langle \widetilde{\mathbf{d}}^g \rangle_w \tag{1}$$

where $\sigma^g$ is the global step size or mutation strength at generation $g$, and $\widetilde{\mathbf{d}}^g$ is considered as a search direction vector.

The path cumulation $\mathbf{s}^{g+1}$ can be detailed as follows [11]:

$$\mathbf{s}^{g+1} = (1 - c_s)\mathbf{s}^g + \sqrt{\mu_{eff}c_s(2 - c_s)}\langle \widetilde{\mathbf{z}}^g \rangle_w \tag{2}$$

where $\langle \widetilde{\mathbf{z}}^g \rangle_w = (\mathbf{C}^g)^{-\frac{1}{2}}\frac{\mathbf{y}^{g+1}-\mathbf{y}^g}{\sigma^g}$. $\mathbf{C}$ is the covariance matrix. The detailed derivation of $\langle \widetilde{\mathbf{z}}^g \rangle_w$ can be found in [11]. $c_s$ can be regarded as memory time constants [12], which is defined as follows [11]:

$$c_s = \frac{\mu_{eff} + 2}{\mu_{eff} + D + 5} \tag{3}$$

where $D$ is the search space dimension, and $\mu_{eff}$ denotes the variance effective selection mass [12], which is calculated as follows [11]:

$$\mu_{eff} = \frac{1}{\sum_{k=1}^{\mu} w_m^2} \tag{4}$$

where $\mu = \frac{\lambda}{2}$, $\lambda = 4 + 3lnD$, $w_m = \frac{1}{\mu}$.

The learning process of the $\mathbf{M}$ matrix instead of the covariance matrix $\mathbf{C}$ of CMA-ES is updated according to the following equation [11]:

$$\mathbf{M}^{g+1} = \mathbf{M}^g[\mathbf{I} + \frac{c_1}{2}(\mathbf{s}^{g+1} \times (\mathbf{s}^{g+1})^{\mathrm{T}} - \mathbf{I}) + \frac{c_w}{2}(\langle \widetilde{\mathbf{z}}^g \times (\widetilde{\mathbf{z}}^g)^{\mathrm{T}} \rangle_w - \mathbf{I})] \tag{5}$$

where I is the identity matrix, $c_w = \min(1 - c_1, \alpha_{cov}\frac{\mu_{eff}+\frac{1}{\mu_{eff}}-2}{(D+2)^2+\frac{\alpha_{cov}\times\mu_{eff}}{2}})$, $c_1 = \frac{\alpha_{cov}}{(D+1.3)^2+\mu_{eff}}$, and $\alpha_{cov} = 2$.

The step-size of the mutation $\sigma^g$ is updated according to the following equation [11]:

$$\sigma^{g+1} = \sigma^g \exp\left[\frac{c_s}{d_\sigma}\left(\frac{\|\mathbf{s}^{g+1}\|}{E[\|\mathcal{N}(0,\mathbf{I})\|]} - 1\right)\right] \tag{6}$$

where $E[\|\mathcal{N}(0,\mathbf{I})\|] = \sqrt{2}\Gamma(\frac{D+1}{2})/\Gamma(\frac{D}{2})$, $d_\sigma = 1 + c_s + 2 \times \max(0, \sqrt{\frac{\mu_{eff}-1}{D+1}} - 1)$.

The pseudo-code of the MA-ES algorithm is shown in Algorithm 1 [11].

---

**Algorithm 1. MA-ES algorithm.**

---

1:    Initialize ($\mathbf{y}^{(0)}$, $\sigma^{(0)}$, $g = 0$, $\mathbf{s}^{(0)} := 0$, $\mathbf{M}^{(0)} = \mathbf{I}$)
2:    **while** termination condition(s) is not fulfilled
3:         **for** $l = 1$ to $\lambda$ **do**
4:              $\widetilde{\mathbf{z}}_l^{(g)} = \mathcal{N}_l(0, \mathbf{I})$
5:              $\widetilde{\mathbf{d}}_l^{(g)} = \mathbf{M}^{(g)}\widetilde{\mathbf{z}}_l^{(g)}$
6:              $\widetilde{f}_l^{(g)} = f(\mathbf{y}^{(g)} + \sigma^{(g)}\widetilde{\mathbf{d}}_l^{(g)})$
7:         **end for**
8:         SortOffspringPopulation
9:         Update $\mathbf{y}^{(g+1)}$ according to (1)
10:        Update $\mathbf{s}^{(g+1)}$ according to (2)
11:        Update $\mathbf{M}^{(g+1)}$ according to (5)
12:        Update $\sigma^{(g+1)}$ according to (6)
13:        $g = g + 1$
14:   **end while**

---

## 4. The MA-ESN-MO Algorithm

Based on the above observations, this section introduces the proposed algorithm MA-ESN-MO. First of all, we introduce the transformation strategy, which transforms an MMOP into an MOP. Then, the MA-ES is employed as a search engine to generate offspring. To obtain solutions with high accuracy, the archive is employed to save better individuals so that the quality of the solutions can be ensured. Finally, the dynamic peak identification strategy is used to enhance the diversity of the population.

### 4.1. Transforming an MMOP into an MOP

The premise of employing multi-objective techniques is that different objectives should conflict with each other. Therefore, two strongly conflicting objectives are constructed as follows:

$$\begin{cases} f_1(\mathbf{x}) = \alpha \times f(\mathbf{x})_{norm} + \mathbf{x} \\ f_2(\mathbf{x}) = \alpha \times f(\mathbf{x})_{norm} - \mathbf{x} \end{cases} \tag{7}$$

where $\alpha$ is a scaling factor that gradually increases during evolution, $f(\mathbf{x})_{norm}$ denotes the normalized objective function values, and $\mathbf{x}$ is the decision variable. An MMOP is transformed into $D$ bi-objective optimization problems, where $D$ is the dimension of an MMOP.

In Equation (7), if the value of $\mathbf{x}$ increases, the value of $-\mathbf{x}$ will decrease, and vice versa. Moreover, $\alpha$ and $f(\mathbf{x})_{norm}$ are positive, which brings the same change for $\mathbf{x}$ and $-\mathbf{x}$. It can be concluded that $f_1(\mathbf{x})$ conflicts with $f_2(\mathbf{x})$. Therefore, the multi-objective optimization methods can be used. $f_1(\mathbf{x})$ and $f_2(\mathbf{x})$ are mainly influenced by $\mathbf{x}$ and $-\mathbf{x}$ at the early stage of evolution. Population distribution can influence the diversity of the population. Gradually, $f_1(\mathbf{x})$ and $f_2(\mathbf{x})$ are mainly influenced by the fitness at the later stage of evolution. Therefore, the algorithm can quickly converge to find the multiple optimal solutions. Similar to the mapping relationship between the decision space and objective space in a MOP, an example of the mapping relationship between an MMOP and an MOP is shown in Figure 1 ($D = 2$). $\alpha$ is designed as follows:

$$\alpha = D \times (b - a) \times \left(\frac{FES}{MaxFES}\right)^D \tag{8}$$

where $D$ denotes the dimension of the problem, $[a, b]^D$ denotes the range of decision space, and *FES* and *MaxFES* denote the number of function evaluations and the maximum number of function evaluations, respectively.

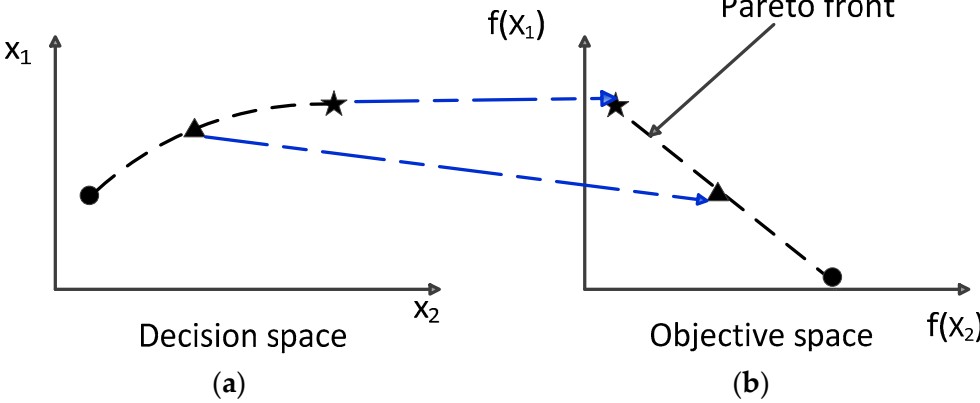

**Figure 1.** Relationship between multimodal optimization in a decision space and multi-objective optimization in an objective space. (**a**) Three optimal solutions of a multimodal function in a decision space. (**b**) Three optimal solutions of a transformed multi-objective function in an objective space.

Figure 2 shows a transformation example of the equal maxima function. The equal maxima function has five global optima. Figure 2a shows the distribution of the population (x) and their fitness (f(x)) when *FES* = 1000. Figure 2b shows the results of the transformation from multimodal optimization to bi-objective optimization. Figure 2c shows the result of the non-dominated sorting procedure, which is employed to find a set of representative Pareto optimal solutions. The representative Pareto optimal solutions will be used as the parents of the next generation. Figure 2d shows the distribution of the individuals and their fitness corresponding to the Pareto optimal solutions.

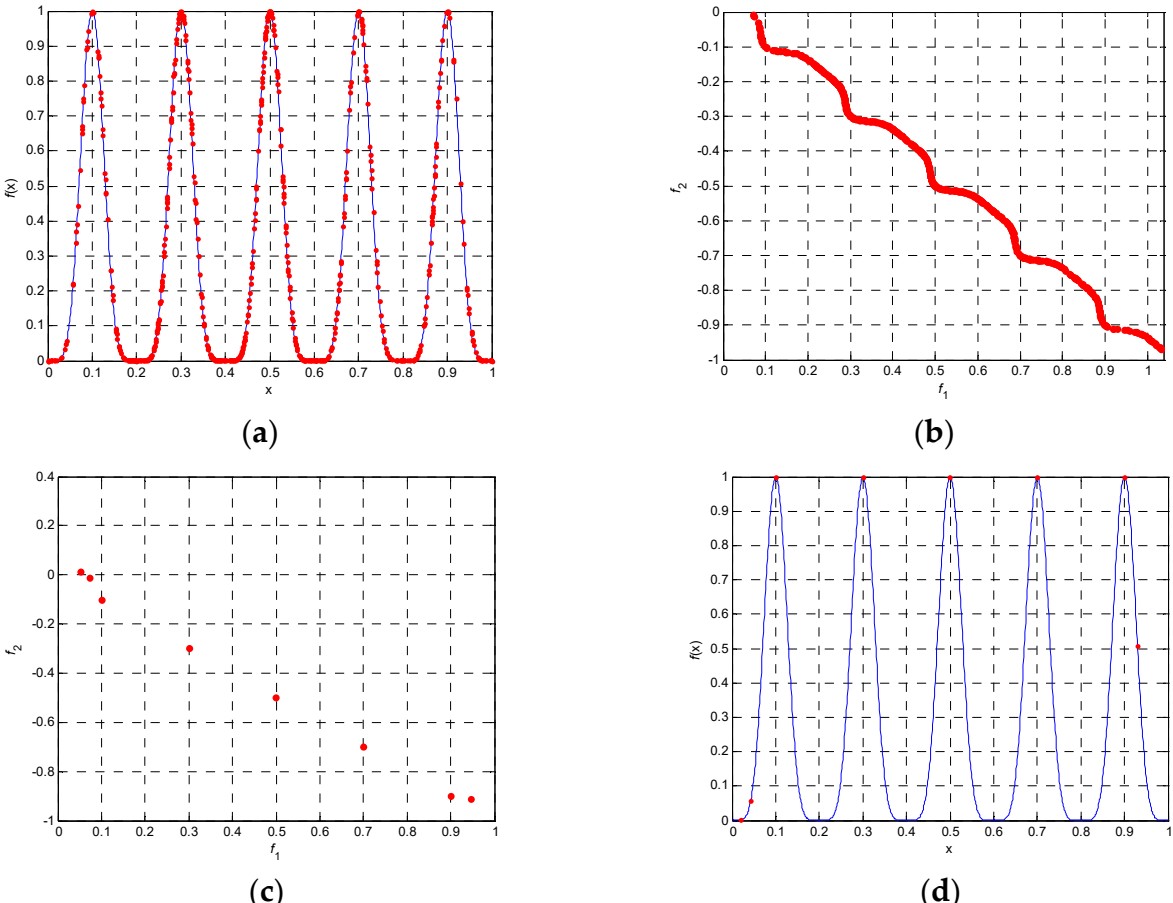

**Figure 2.** Transformation example of the equal maxima function.

*4.2. Matrix Adaptation Evolution Strategy with Multi-Objective Optimization Algorithm*

The exploration efficiency is dependent on the distribution of individuals, namely the population diversity. The exploitation efficiency is associated with the fitness of individuals. Then, diversity and fitness can be used to indirectly measure the exploration and exploitation, respectively. Therefore, diversity and fitness can be employed to achieve a trade-off between exploration and exploitation [39]. In view of this idea, the diversity and the fitness are considered in Equation (7). In the proposed algorithm, the non-dominated sorting mechanism is used to find the non-dominated solutions that are helpful for exploring the solution space efficiently and exhaustively. The non-dominated solutions, which take into account fitness and diversity, can be used as the seeds of niching. The non-dominated sorting procedure comes from the non-dominated sorting genetic algorithm II (NSGA-II) [40], which is a classical multi-objective optimization algorithm. Details of the fast non-dominated sorting algorithm can be found in [40].

In MA-ES, the offspring is yielded by the parent that is mutated according to a normal distribution. Then the parent will be discarded, while the excellent offspring will be preserved as the parent for the next generation. However, sometimes, the offspring may achieve worse performance than their

parents. Take $f_2$ equal maxima (1D) as an example; $f_2$ has five global optima. Figure 3 shows three independent runs of MA-ES on $f_2$ (the equal maxima 1D function). It can be seen that the number of global optima is not stable during the evolution. Originally, the parents have found the global optima. However, the parents are not preserved. The offspring are only near the optimal solution. Therefore, the number of the global optima found by the algorithm is variable during the evolution. In order to solve the problem, the archive is employed to save the individuals that have better performance. Specifically speaking, if the offspring performs worse, the individuals in the archive will be used as the parents for the next generation.

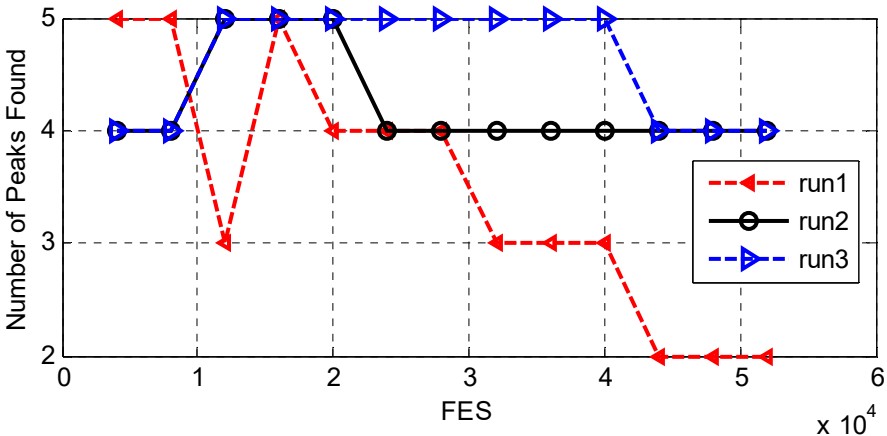

**Figure 3.** Number of global optima found by the matrix adaptation evolution strategy (MA-ES) over three independent runs on $f_2$.

Based on the above explanation, the pseudo-code of the MA-ESN-MO algorithm is illustrated in Algorithm 2, from which we can see that the MA-ESN-MO generation cycle is performed within the while-end while-loop (line 3 to line 26). $\mathbf{C}^{(g)}$ denotes the number of global optima found at generation $g$. At each generation, a number of $\lambda$ offspring is generated according to Equation (1), where the parental stage $\mathbf{x}^g$ is mutated according to a normal distribution yielding offspring $\mathbf{y}^g$. For a given individual $i$, $f(\mathbf{y}_i)$ can be calculated from the original objective function of MMOP. Then, the population is sorted according to the individual fitness in line (6). Afterward, the two objective values of the transformed bi-objective optimization problem can be obtained according to Equations (7) and (8). Then, the seeds of niching (also referred to as the parent of the next generation) are generated by the non-dominated sorting procedure or dynamic peak identification algorithm according to the condition in line (8). Then, the path cumulation $s$, matrix $\mathbf{M}$, and the step size $\sigma$ (also referred to as mutation strength) are updated according to Equations (2), (5) and (6), respectively. At each generation, the excellent individual will be preserved in the archive as the parent for the next generation (line 18, line 23). Therefore, if $\mathbf{C}^{(g)}$ is less than $\mathbf{C}^{(g-1)}$, this shows that the performance of offspring is worse than that of their parents. Then, the individuals in the archive will be used as the parents for the next generation (line 21).

---

**Algorithm 2. MA-ESN-MO algorithm.**

---

1:　Initialize $D$(number of dimensions), $\lambda$, $\mathbf{y}^{(0)}$, $\boldsymbol{\sigma}^{(0)}$, $\mathbf{s}^{(0)}$, $\mathbf{d}^{(0)}$, $\mathrm{M}^{(0)}$ and $NP$, $g = 0$
2:　Initialize $NP$ parents
3:　**while** the termination condition is not satisfied
4:　　　Generate the individuals according to Equation (1)
5:　　　Calculate the objective function value of each individual
6:　　　Sort the population according to the objective function value
7:　　　Compute the $f_1$ and $f_2$ values of each dimension according to equations (7) and (8)
8:　　　**if** *rand* > 0.5
9:　　　　Generate the seeds of niching according to the non-dominated sorting procedure
10:　　　**else**
11:　　　　Generate the seeds of niching with the **dynamic peak identification algorithm** (Algorithm 2)
12:　　　**end if**
13:　　　Update $\mathbf{s}^{(g+1)}$ according to Equation (2)
14:　　　Update $\mathbf{M}^{(g+1)}$ according to Equation (5)
15:　　　Update $\boldsymbol{\sigma}^{(g+1)}$ according to Equation (6)
16:　　　$g = g + 1$
17:　　　**if** $g == 1$
18:　　　　Archive = $\mathbf{x}^{(g)}$
19:　　　**else**
20:　　　　**if** $\mathbf{C}^{(g)} < \mathbf{C}^{(g-1)}$
21:　　　　　$\mathbf{x}^{(g)}$ = Archive
22:　　　　**else**
23:　　　　　Archive = $\mathbf{x}^{(g)}$
24:　　　　**end if**
25:　　　**end if**
26:　**end while**
**Output: the global optima with the maximum objective function value in the population.**

---

The pseudo-code of dynamic peak identification is presented in Algorithm 3. Firstly, the population is sorted according to the objective function value. For a given individual *Pop*{*i*}, using an estimated so-called niche radius $\rho$, *Pop*{*i*} is classified into a peak and populates this niche (line 12 to line 15). In addition, every niche includes several individuals (line 7 to line 11). Then, the various fitness peaks are identified dynamically within the for-end for-loop (line 6 to line 16).

---

**Algorithm 3. Dynamic Peak Identification [15].**

---

1:　Input niche radius $\rho$, population *Pop*, and population size $NP$
2:　Sort *Pop* according to the objective function value
3:　NumPeak = 1
4:　DPS = {*Pop*{1}}
5:　Niche (NumPeak) = {*Pop*{1}}
6:　**for** $i$ = 2 to $NP$
7:　　**for** $k$ = 1 to NumPeak
8:　　　**if** *Pop*{*i*} and DPS (*k*) belong to the same niche
9:　　　　Niche (*k*) = Niche (*k*)∪{*Pop*{*i*}}
10:　　　**end if**
11:　　**end for**
12:　　**if** *Pop*{*i*} is not within $\rho$ of peak in DPS
13:　　　NumPeak = NumPeak + 1
14:　　　DPS = DPS∪{*Pop*{*i*}}
15:　　**end if**
16:　**end for**
**Output: DPS and Niche**

---

As mentioned earlier, MA-ESN-MO is proposed by introducing three improvement strategies to solve multimodal optimization problems: a transformation strategy, an archive strategy, and a dynamic peak identification strategy. The transformation strategy is used to transform an MMOP into an MOP. The advantage of the transformation strategy is that it is unnecessary to use the problem-dependent niching parameters. The archive is used to save better individuals, which is effective in preserving the seeds of niching. The dynamic peak identification strategy is employed to prevent the best niche from occupying the population's resources.

There are three main differences among the MA-ESN-MO and the other algorithms mentioned before [4,15–19].

(1)　Two conflicting objective functions are constructed differently. The landscape information of the population is usually helpful for judging the diversity of the population. The fitness of the objective function is helpful for obtaining the global optimum. Therefore, the two strongly conflicting objectives are designed by the landscape information of the population and the fitness of the objective function. In order to find multiple optimal solutions, exploration should be paid attention to in the early stage of evolution, and exploitation should be considered in the later stage of evolution. Therefore, the parameter $\alpha$, which is proportional to the number of function evaluations, is introduced to adjust for exploration and exploitation.

(2)　The archive is employed to save the seeds of niching. In the classical CMA-ES algorithm and MA-ES algorithm, the parent will be discarded after the offspring are yielded. The CMA-ES and MA-ES perform well in solving unimodal optimization problems because of their strong exploratory ability. In multimodal optimization problems, the parent is responsible for finding the area where the potential optimal solution existed, while the offspring are responsible for exploitation. However, sometimes the offspring perform worse than their parents. As a result, the offspring may fail to find the optimal solution. As the evolution continues, the area where the potential optimal solution existed may be abandoned. Therefore, it will be difficult for the population to find all the global optimal solutions. To alleviate this issue, the archive is introduced in the MA-ESN-MO to save better individuals. At each generation, the individuals in the archive will be used as the parents for the next generation. If the optimal individual from offspring performs better than its parent, the individual will be saved in the archive. Otherwise, the parent will be saved in the archive. Then, it ensures the population to find all the global optimal solutions.

(3)　The non-dominated solutions include all the multiple optima of an MMOP. However, they also include some inferior solutions. Then, the best niche may occupy the population's resources during the evolution. To address this issue, the dynamic peak identification strategy is used to avoid converging toward a single global optimum. However, the performance of the dynamic peak identification strategy is highly sensitive to the niching radius. In other words, the performance of the algorithm deteriorates with an inappropriate niching radius. Therefore, the transformation strategy and the dynamic peak identification strategy are dynamically employed in MA-ESN-MO to improve the performance of the algorithm.

## 5. Experiments and Discussions

To verify the effectiveness of the improvement strategies proposed in this paper, 20 test problems from CEC2013 [21] multimodal benchmarks are used. The algorithms for testing include CMA-ES [4], MA-ES [11], CMA-N [14], CMA-NMO, CMA-NMM [20], CMA-NMM-MO, MA-ESN [11,15], and MA-ESN-MO. CMA-NMO, CMA-NMM-MO, and MA-ESN-MO are produced by introducing the improvement strategies into CMA-ES, CMA-NMM, and MA-ES, respectively. To obtain an unbiased comparison, all the experiments are run on a PC with an Intel Core i7-3770 3.40 GHz CPU and 4 GB memory. All of the experiments are run 25 times, and the codes are implemented in Matlab R2013a (MathWorks, Natick, MA, USA).

### 5.1. Parameter Settings and Performance Criteria

Table 1 shows a brief description of testing problems. $f_1$–$f_{10}$ are simple and low-dimensional multimodal problems, while $f_{11}$–$f_{20}$ are composition multimodal problems composed of several basic problems with different characteristics. $f_1$–$f_5$ have a small number of global optima, and $f_6$–$f_{10}$ have a large number of global optima. For each algorithm, the subpopulation size is set to 10. Some functions from the CEC2013 are drawn in the following. The equal maxima ($f_2$) has five global optima. There are no local optima, as shown in Figure 4. The Himmelblau function ($f_4$) has four global optima, as shown in Figure 5. Figure 6 shows an example of the Shubert 2D function ($f_6$), where there are 18 global optima in nine pairs. Figure 7 shows the 2D version of Composition Function 2 (CF$_2$). CF$_2$ ($f_{12}$) is constructed based on eight basic functions ($n = 8$), thus it has eight global optima.

**Table 1.** Parameter setting for test functions.

| Fun. | $\varepsilon$ | $r$ | D | Number of Global Optima | Number of Function Evaluations | Population Size |
|---|---|---|---|---|---|---|
| $f_1$ | 0.1/0.01/0.001/0.0001/0.00001 | 0.01 | 1 | 2 | $5 \times 10^4$ | 80 |
| $f_2$ | 0.1/0.01/0.001/0.0001/0.00001 | 0.01 | 1 | 5 | $5 \times 10^4$ | 80 |
| $f_3$ | 0.1/0.01/0.001/0.0001/0.00001 | 0.01 | 1 | 1 | $5 \times 10^4$ | 80 |
| $f_4$ | 0.1/0.01/0.001/0.0001/0.00001 | 0.01 | 2 | 4 | $5 \times 10^4$ | 80 |
| $f_5$ | 0.1/0.01/0.001/0.0001/0.00001 | 0.5 | 2 | 2 | $5 \times 10^4$ | 80 |
| $f_6$ | 0.1/0.01/0.001/0.0001/0.00001 | 0.5 | 2 | 18 | $2 \times 10^5$ | 100 |
| $f_7$ | 0.1/0.01/0.001/0.0001/0.00001 | 0.2 | 2 | 36 | $2 \times 10^5$ | 300 |
| $f_8$ | 0.1/0.01/0.001/0.0001/0.00001 | 0.5 | 3 | 81 | $4 \times 10^5$ | 300 |
| $f_9$ | 0.1/0.01/0.001/0.0001/0.00001 | 0.2 | 3 | 216 | $4 \times 10^5$ | 300 |
| $f_{10}$ | 0.1/0.01/0.001/0.0001/0.00001 | 0.01 | 2 | 12 | $2 \times 10^5$ | 100 |
| $f_{11}$ | 0.1/0.01/0.001/0.0001/0.00001 | 0.01 | 2 | 6 | $2 \times 10^5$ | 200 |
| $f_{12}$ | 0.1/0.01/0.001/0.0001/0.00001 | 0.01 | 2 | 8 | $2 \times 10^5$ | 200 |
| $f_{13}$ | 0.1/0.01/0.001/0.0001/0.00001 | 0.01 | 2 | 6 | $2 \times 10^5$ | 200 |
| $f_{14}$ | 0.1/0.01/0.001/0.0001/0.00001 | 0.01 | 3 | 6 | $4 \times 10^5$ | 200 |
| $f_{15}$ | 0.1/0.01/0.001/0.0001/0.00001 | 0.01 | 3 | 8 | $4 \times 10^5$ | 200 |
| $f_{16}$ | 0.1/0.01/0.001/0.0001/0.00001 | 0.01 | 5 | 6 | $4 \times 10^5$ | 200 |
| $f_{17}$ | 0.1/0.01/0.001/0.0001/0.00001 | 0.01 | 5 | 8 | $4 \times 10^5$ | 200 |
| $f_{18}$ | 0.1/0.01/0.001/0.0001/0.00001 | 0.01 | 10 | 6 | $4 \times 10^5$ | 200 |
| $f_{19}$ | 0.1/0.01/0.001/0.0001/0.00001 | 0.01 | 10 | 8 | $4 \times 10^5$ | 200 |
| $f_{20}$ | 0.1/0.01/0.001/0.0001/0.00001 | 0.01 | 20 | 8 | $4 \times 10^5$ | 200 |

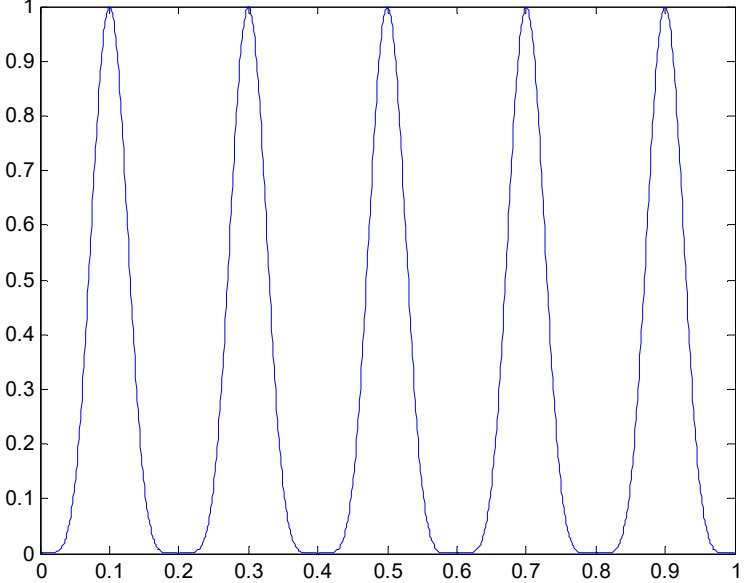

**Figure 4.** Equal maxima.

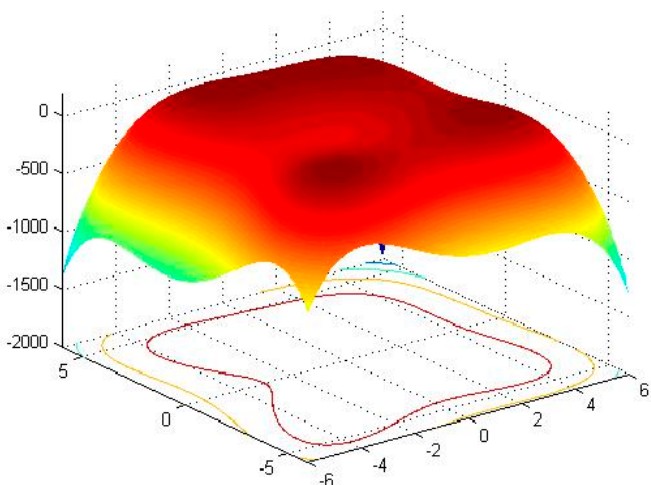

**Figure 5.** Himmelblau.

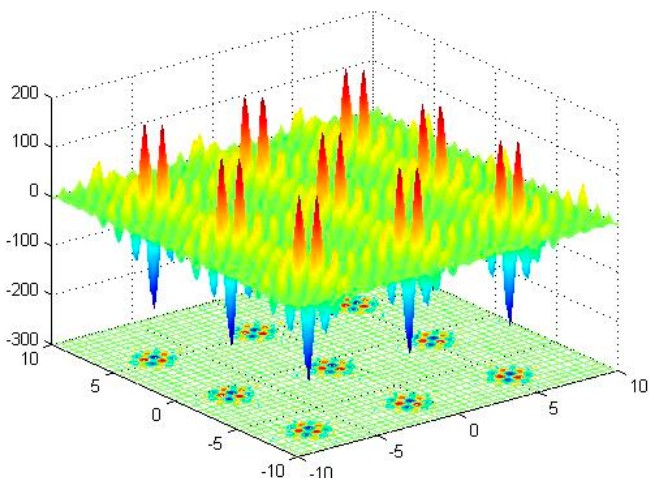

**Figure 6.** Shubert 2D function.

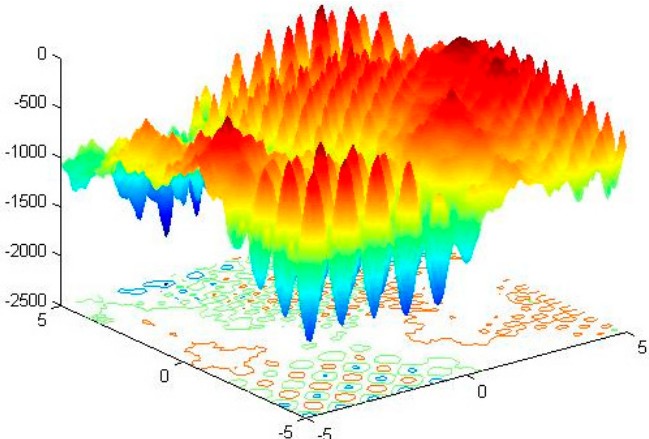

**Figure 7.** Composition Function 2.

In this experiment, three criteria [6] are employed to measure the performance of different multimodal optimization algorithms on each function.

### 5.1.1. Peak Ratio

The peak ratio (*PR*) is used to measure the average percentage of all known global optima found over *NR* independent runs:

$$PR = \frac{\sum_{run=1}^{NR} NGF_{run}}{NGO * NR} \tag{9}$$

where $NGF_{run}$ is the number of global optima found in the *run*-th run, *NGO* is the number of true global optima, and *NR* is the number of runs (*NR* = 25).

### 5.1.2. Success Rate

The success rate denotes the percentage of successfully detecting all the global optima out of NR runs for each function.

### 5.1.3. Average Number of Peaks Found

The average number of peaks (*ANP*) denotes the average number of peaks found by an algorithm over NR runs.

$$ANP = \frac{\sum_{run=1}^{NR} NGF_{run}}{NR} \tag{10}$$

### *5.2. Experimental Results of Eight Optimization Algorithms*

The experimental results and analyses are shown in the following. The best solution of every single function in different algorithms is highlighted with boldface. The results of the test functions from $f_1$ to $f_{10}$ are shown in Table 2. As can be seen, the performance of the CMA-ES is the same as MA-ES on $f_2$–$f_5$, $f_7$, and f$_{10}$. The CMA-ES performs slightly better than the MA-ES on $f_1$, $f_6$, $f_8$, and $f_9$. As mentioned earlier, the MA-ES is a simplified version of the CMA-ES; therefore, their performance is similar. The CMA-NMO performs better than the CMA-N on $f_2$, $f_4$, $f_5$, $f_6$, $f_8$, and $f_{10}$. The CMA-N beats the CMA-NMO on $f_7$ and $f_9$ when $\varepsilon = 0.1$. The CMA-NMM-MO performs better than the CMA-NMM on $f_4$–$f_{10}$. The CMA-NMM outperforms the CMA-NMM-MO on $f_2$ and $f_3$. The MA-ESN-MO performs better than the MA-ESN on $f_2$–$f_6$, $f_8$, and $f_{10}$. The MA-ESN beats the MA-ESN-MO on $f_7$ and $f_9$ when $\varepsilon = 0.1$. The experimental results of the test functions from $f_1$–$f_{10}$ show that the performance of the CMA-NMO, CMA-NMM-MO, and MA-ESN-MO has been greatly improved because of the introduction of the multi-objective optimization strategy, the dynamic peak identification strategy, and the archive.

Table 3 shows the results of the test function from $f_{11}$ to $f_{20}$. As can be seen, the CMA-ES performs slightly better than the MA-ES. This suggests that the two evolution paths are more effective in solving complex problems. The CMA-NMO performs better than the CAM-N on $f_{12}$, but worse on $f_{20}$. For functions $f_{11}$–$f_{19}$, the CMA-NMO performs better than the CMA-N on different $\varepsilon$ except for $\varepsilon = 0.1$. For function $f_{11}$–$f_{20}$, the CMA-NMM-MO performs better than the CMA-NMM except for function $f_{19}$ and $f_{20}$. The MA-ESN-MO performs better than the MA-ESN on $f_{12}$ but worse on $f_{19}$ and $f_{20}$ than the MA-ESN. For functions $f_{11}$, $f_{13}$–$f_{18}$, the MA-ESN-MO performs better than the MA-ESN on different $\varepsilon$ except $\varepsilon = 0.1$. The experimental results show that the CMA-NMO, CMA-NMM-MO, and MA-ESN-MO can find better solutions than the CMA-ES, MA-ES, CMA-N, CMA-NMM, and MA-ESN.

Tables 4 and 5 shows the results of ANP in terms of the mean value (Mean), Peak ratio (*PR*), and Success Rate (*SR*) obtained in 25 independent runs by each algorithm for functions $f_1$ to $f_{10}$ and $f_{11}$ to $f_{20}$, respectively. In view of statistics, the Wilcoxon signed-rank test [41] at the 5% significance level is used to compare the MA-ESN-MO with other compared algorithms. "≈", "+", and "−" are applied to express the performance of the MA-ESN-MO as similar to, worse than, and better than that of the compared algorithm, respectively. The statistical results are reported in Tables 4 and 5. Table 4 shows that all the algorithms can accurately find multiple optimal solutions on $f_1$ except for the MA-ES. The CMA-NMM performs better than the other algorithms on function $f_2$. For function $f_3$, all of the algorithms can accurately find optimal solutions except for CMA-N and

MA-ESN. The CMA-NMO, CMA-NMM-MO, and MA-ESN-MO perform better than other algorithms on $f_4$–$f_{10}$. For $f_1$–$f_{10}$, the MA-ESN-MO works better than the CMA-ES, MA-ES, CMA-N, CMA-NMO, CMA-NMM, CMA-NMM-MO, and MA-ESN on eight, eight, eight, zero, seven, zero, and eight test problems, respectively. Table 5 shows that the CMA-NMO, CMA-NMM-MO, and MA-ESN-MO perform better than other algorithms on $f_{11}$–$f_{17}$. The CMA-NMO performs the best on $f_{18}$. For function $f_{19}$, the performance of all the algorithms is basically the same except for the CMA-ES and MA-ES. None of the algorithms can find multiple optimal solutions on $f_{20}$. For $f_{11}$–$f_{20}$, the MA-ESN-MO works better than the CMA-ES, MA-ES, CMA-N, CMA-NMO, CMA-NMM, CMA-NMM-MO, and MA-ESN on nine, nine, seven, zero, seven, one, and seven test problems, respectively.

**Table 2.** Experimental results of *ANP* (average peaks found) obtained by the covariance matrix adaptation evolution strategy (CMA-ES), matrix adaptation evolution strategy (MA-ES), CMA-ES-Niching (CMA-N) CMA-ES-Niching with the multi-objective optimization algorithm (CMA-NMO), CMA-ES Niching with Mahalanobis Metric (CMA-NMM), CMA-NMM with multi-objective optimization algorithm (CMA-NMM-MO), MA-ES-Niching (MA-ESN), and matrix adaptation evolution strategy with multi-objective optimization algorithm (MA-ESN-MO) for functions $f_1$–$f_{10}$.

| Fun. | $\varepsilon$ | CMA-ES | MA-ES | CMA-N | CMA-NMO | CMA-NMM | CMA-NMM-MO | MA-ESN | MA-ESN-MO |
|---|---|---|---|---|---|---|---|---|---|
| $f_1$ | 0.1 | 2 | 2 | 2 | 2 | 2 | 2 | 2 | 2 |
| | 0.01 | 2 | 2 | 2 | 2 | 2 | 2 | 2 | 2 |
| | 0.001 | 2 | 2 | 2 | 2 | 2 | 2 | 2 | 2 |
| | 0.00001 | 2 | 1.96 | 2 | 2 | 2 | 2 | 2 | 2 |
| $f_2$ | 0.1 | 1 | 1 | 5 | 5 | 5 | 5 | 5 | 5 |
| | 0.01 | 1 | 1 | 5 | 5 | 5 | 5 | 4.92 | 5 |
| | 0.001 | 1 | 1 | 4.52 | 5 | 5 | 5 | 4.32 | 5 |
| | 0.00001 | 1 | 1 | 3.20 | 3.68 | 5 | 1.96 | 2.16 | 1.44 |
| $f_3$ | 0.1 | 1 | 1 | 1 | 1 | 1 | 1 | 1 | 1 |
| | 0.01 | 1 | 1 | 1 | 1 | 1 | 1 | 1 | 1 |
| | 0.001 | 1 | 1 | 1 | 1 | 1 | 1 | 0.96 | 1 |
| | 0.00001 | 1 | 1 | 0.60 | 0.28 | 1 | 0.48 | 0.56 | 0.60 |
| $f_4$ | 0.1 | 1 | 1 | 4 | 4 | 2.68 | 4 | 4 | 4 |
| | 0.01 | 1 | 1 | 4 | 4 | 2.68 | 4 | 4 | 4 |
| | 0.001 | 1 | 1 | 2.64 | 4 | 2.68 | 4 | 2.56 | 4 |
| | 0.00001 | 1 | 1 | 2.56 | 3.96 | 2.56 | 3.80 | 2.36 | 3.96 |
| $f_5$ | 0.1 | 1.12 | 1.04 | 2 | 2 | 2 | 2 | 2 | 2 |
| | 0.01 | 1.12 | 1.04 | 1.80 | 2 | 1.64 | 2 | 1.80 | 2 |
| | 0.001 | 1.12 | 1.04 | 1.52 | 2 | 1.48 | 2 | 1.60 | 2 |
| | 0.00001 | 1.12 | 1.04 | 1.40 | 1.80 | 1.32 | 1.72 | 1.32 | 1.80 |
| $f_6$ | 0.1 | 1 | 1 | 13.56 | 17.56 | 12.72 | 17.52 | 14.12 | 17.40 |
| | 0.01 | 1 | 1 | 13.48 | 17.48 | 12.72 | 17.48 | 14.04 | 17.36 |
| | 0.001 | 1 | 1 | 13.44 | 17.40 | 12.72 | 17.36 | 14.00 | 17.36 |
| | 0.00001 | 1 | 0.96 | 13.40 | 17.20 | 12.72 | 17.24 | 14.00 | 17.04 |
| $f_7$ | 0.1 | 1 | 1 | 36.00 | 35.28 | 27.68 | 35.04 | 36.00 | 35.20 |
| | 0.01 | 1 | 1 | 29.68 | 30.44 | 27.68 | 29.00 | 28.84 | 30.00 |
| | 0.001 | 1 | 1 | 29.40 | 30.28 | 27.68 | 28.96 | 28.84 | 29.64 |
| | 0.00001 | 1 | 1 | 29.52 | 29.92 | 27.16 | 28.32 | 28.56 | 29.12 |
| $f_8$ | 0.1 | 1.08 | 0.96 | 42.64 | 53.40 | 44.04 | 54.04 | 43.44 | 50.84 |
| | 0.01 | 1.08 | 0.92 | 42.64 | 52.96 | 44.04 | 53.48 | 43.40 | 50.20 |
| | 0.001 | 1.08 | 0.76 | 42.64 | 52.40 | 44.04 | 52.92 | 43.40 | 49.68 |
| | 0.00001 | 1.08 | 0.64 | 42.60 | 50.80 | 44.04 | 51.28 | 43.36 | 47.72 |
| $f_9$ | 0.1 | 1 | 1.04 | 216 | 89.20 | 28.92 | 73.32 | 216 | 81.48 |
| | 0.01 | 1 | 1.00 | 31.40 | 57.32 | 28.92 | 52.16 | 30.48 | 54.40 |
| | 0.001 | 1 | 1.00 | 31.40 | 46.92 | 28.92 | 41.60 | 30.48 | 43.72 |
| | 0.00001 | 1 | 0.96 | 31.40 | 33.56 | 28.92 | 31.00 | 30.48 | 31.00 |
| $f_{10}$ | 0.1 | 1 | 1 | 9.92 | 11.68 | 8.64 | 11.72 | 9.68 | 11.72 |
| | 0.01 | 1 | 1 | 9.92 | 11.60 | 8.64 | 11.72 | 9.68 | 11.56 |
| | 0.001 | 1 | 1 | 9.92 | 11.40 | 8.64 | 11.64 | 9.68 | 11.28 |
| | 0.00001 | 1 | 1 | 9.92 | 11.32 | 8.64 | 11.24 | 9.68 | 10.72 |

**Table 3.** Experimental results of ANP (average peaks found) obtained by CMA-ES, MA-ES, CMA-N, CMA-NMO, CMA-NMM, CMA-NMM-MO, MA-ESN, and MA-ESN-MO for functions $f_{11}$–$f_{20}$.

| Fun. | $\varepsilon$ | CMA-ES | MA-ES | CMA-N | CMA-NMO | CMA-NMM | CMA-NMM-MO | MA-ESN | MA-ESN-MO |
|---|---|---|---|---|---|---|---|---|---|
| | 0.1 | 1 | 1.04 | 6.00 | 4.28 | 3.72 | 4.16 | 6.00 | 4.04 |
| | 0.01 | 1 | 1.04 | 3.68 | 3.96 | 3.72 | 3.96 | 3.72 | 4.00 |
| $f_{11}$ | 0.001 | 1 | 1.04 | 3.68 | 3.92 | 3.72 | 3.96 | 3.60 | 4.00 |
| | 0.00001 | 1 | 1.04 | 3.56 | 3.88 | 3.72 | 3.92 | 3.36 | 3.96 |
| | 0.1 | 1 | 1 | 2.84 | 6.92 | 2.80 | 6.24 | 3.00 | 7.04 |
| | 0.01 | 1 | 1 | 2.76 | 6.44 | 2.80 | 5.60 | 2.68 | 6.60 |
| $f_{12}$ | 0.001 | 1 | 1 | 2.64 | 6.08 | 2.80 | 5.40 | 2.48 | 6.08 |
| | 0.00001 | 1 | 1 | 2.64 | 5.80 | 2.80 | 5.36 | 2.40 | 5.80 |
| | 0.1 | 1 | 1 | 5.76 | 3.92 | 3.44 | 3.96 | 5.64 | 3.92 |
| | 0.01 | 1 | 1 | 3.36 | 3.84 | 3.44 | 3.96 | 3.44 | 3.92 |
| $f_{13}$ | 0.001 | 1 | 1 | 3.32 | 3.84 | 3.44 | 3.96 | 3.32 | 3.92 |
| | 0.00001 | 1 | 1 | 3.16 | 3.72 | 3.24 | 3.92 | 3.20 | 3.92 |
| | 0.1 | 1 | 0.96 | 6.00 | 3.56 | 1.72 | 3.68 | 5.80 | 3.64 |
| | 0.01 | 1 | 0.76 | 1.76 | 3.52 | 1.72 | 3.68 | 1.84 | 3.60 |
| $f_{14}$ | 0.001 | 1 | 0.76 | 1.76 | 3.52 | 1.72 | 3.68 | 1.84 | 3.52 |
| | 0.00001 | 1 | 0.72 | 1.76 | 3.48 | 1.72 | 3.48 | 1.84 | 3.48 |
| | 0.1 | 1 | 2.08 | 8.00 | 2.40 | 1.12 | 2.44 | 7.44 | 2.20 |
| | 0.01 | 1 | 0.84 | 1.20 | 2.24 | 1.12 | 2.44 | 1.32 | 2.12 |
| $f_{15}$ | 0.001 | 1 | 0.80 | 1.20 | 2.08 | 1.12 | 2.44 | 1.32 | 2.12 |
| | 0.00001 | 1 | 0.72 | 1.20 | 2.00 | 1.12 | 2.40 | 1.32 | 2.12 |
| | 0.1 | 1 | 0 | 6.00 | 2.28 | 1.16 | 2.72 | 6.00 | 2.32 |
| | 0.01 | 1 | 0 | 1.24 | 2.16 | 1.16 | 2.44 | 1.08 | 2.08 |
| $f_{16}$ | 0.001 | 1 | 0 | 1.24 | 2.04 | 1.16 | 2.28 | 1.08 | 2.00 |
| | 0.00001 | 1 | 0 | 1.24 | 1.92 | 1.16 | 2.24 | 1.08 | 1.84 |
| | 0.1 | 0.84 | 0 | 6.32 | 1.72 | 1 | 1.52 | 5.92 | 1.56 |
| | 0.01 | 0.84 | 0 | 1.00 | 1.52 | 1 | 1.44 | 1.04 | 1.56 |
| $f_{17}$ | 0.001 | 0.84 | 0 | 1.00 | 1.52 | 1 | 1.44 | 1.04 | 1.56 |
| | 0.00001 | 0.84 | 0 | 1.00 | 1.48 | 1 | 1.40 | 1.04 | 1.48 |
| | 0.1 | 3.88 | 0 | 6 | 1.84 | 2.32 | 1.52 | 6 | 1.60 |
| | 0.01 | 0.88 | 0 | 1 | 1.60 | 1.00 | 1.48 | 1 | 1.60 |
| $f_{18}$ | 0.001 | 0.24 | 0 | 1 | 1.56 | 1.00 | 1.48 | 1 | 1.56 |
| | 0.00001 | 0.16 | 0 | 1 | 1.16 | 1.00 | 1.08 | 1 | 1.04 |
| | 0.1 | 0.72 | 0 | 1.20 | 1.12 | 1.08 | 1.04 | 1.40 | 1.08 |
| | 0.01 | 0.20 | 0 | 1.00 | 1.12 | 1.00 | 1.00 | 1.00 | 1.04 |
| $f_{19}$ | 0.001 | 0.16 | 0 | 1.00 | 1.08 | 1.00 | 0.84 | 1.00 | 0.96 |
| | 0.00001 | 0 | 0 | 1.00 | 1.00 | 1.00 | 0.80 | 1.00 | 0.80 |
| | 0.1 | 0 | 0 | 8 | 1 | 7.72 | 1 | 8 | 0.92 |
| | 0.01 | 0 | 0 | 0.88 | 0.16 | 0.96 | 0.36 | 0.56 | 0 |
| $f_{20}$ | 0.001 | 0 | 0 | 0.04 | 0 | 0 | 0 | 0 | 0 |
| | 0.00001 | 0 | 0 | 0 | 0 | 0 | 0 | 0 | 0 |

**Table 4.** Experimental results of ANP, Peak ratio (PR) and Success Rate (SR) obtained by the CMA-ES, MA-ES, CMA-N, CMA-NMO, CMA-NMM, CMA-NMM-MO, MA-ESN, and MA-ESN-MO on $\varepsilon = 0.0001$ for functions $f_1$ to $f_{10}$.

| | Fun. | CMA-ES | MA-ES | CMA-N | CMA-NMO | CMA-NMM | CMA-NMM-MO | MA-ESN | MA-ESN-MO |
|---|---|---|---|---|---|---|---|---|---|
| | ANP | 2($\approx$) | 1.96($\approx$) | 2($\approx$) | 2($\approx$) | 2($\approx$) | 2($\approx$) | 2($\approx$) | 2 |
| $f_1$ | PR | 1 | 0.98 | 1 | 1 | 1 | 1 | 1 | 1 |
| | SR | 100% | 96% | 100% | 100% | 100% | 100% | 100% | 100% |
| | ANP | 1(−) | 1(−) | 3.64(−) | 4.32($\approx$) | 5(+) | 4.28($\approx$) | 3.04(−) | 4.40 |
| $f_2$ | PR | 0.2 | 0.2 | 0.72 | 0.86 | 1 | 8.56 | 0.68 | 0.88 |
| | SR | 0% | 0% | 28% | 40% | 100% | 40% | 4% | 44% |
| | ANP | 1($\approx$) | 1($\approx$) | 0.72(−) | 1($\approx$) | 1($\approx$) | 1($\approx$) | 0.72(−) | 1 |
| $f_3$ | PR | 1 | 1 | 0.72 | 1 | 1 | 1 | 0.72 | 1 |
| | SR | 100% | 100% | 72% | 100% | 100% | 100% | 72% | 100% |
| | ANP | 1(−) | 1(−) | 2.60(−) | 3.96($\approx$) | 2.56(−) | 4($\approx$) | 2.48(−) | 4 |
| $f_4$ | PR | 0.25 | 0.25 | 0.65 | 0.99 | 0.64 | 1 | 0.62 | 1 |
| | SR | 0% | 0% | 8% | 96% | 8% | 100% | 4% | 100% |
| | ANP | 1.12(−) | 1.04(−) | 1.52(−) | 1.92($\approx$) | 1.36(−) | 1.92($\approx$) | 1.48(−) | 1.92 |
| $f_5$ | PR | 0.56 | 0.52 | 0.76 | 0.96 | 0.68 | 0.96 | 0.74 | 0.96 |
| | SR | 12% | 4% | 60% | 92% | 48% | 92% | 52% | 92% |
| | ANP | 1(−) | 1(−) | 13.40(−) | 17.36($\approx$) | 12.72(−) | 17.36($\approx$) | 14.00(−) | 17.20 |
| $f_6$ | PR | 0.05 | 0.05 | 0.74 | 0.96 | 0.70 | 0.96 | 0.77 | 0.95 |
| | SR | 0% | 0% | 0% | 52% | 4% | 60% | 0% | 44% |

**Table 4.** *Cont.*

| Fun. | | CMA-ES | MA-ES | CMA-N | CMA-NMO | CMA-NMM | CMA-NMM-MO | MA-ESN | MA-ESN-MO |
|------|-----|--------|-------|-------|---------|---------|-----------|--------|-----------|
| $f_7$ | ANP | 1(−) | 1(−) | 29.52(≈) | 30.00(≈) | 27.60(−) | 28.72(≈) | 28.60(≈) | 29.20 |
| | PR | 0.02 | 0.02 | 0.82 | 0.83 | 0.76 | 0.79 | 0.79 | 0.82 |
| | SR | 0% | 0% | 0% | 0% | 0% | 0% | 0% | 0% |
| $f_8$ | ANP | 1.08(−) | 0.68(−) | 42.60(−) | 51.60(≈) | 44.04(−) | 52.04(≈) | 43.36(−) | 48.88 |
| | PR | 0.01 | 0.01 | 0.52 | 0.63 | 0.54 | 0.64 | 0.53 | 0.60 |
| | SR | 0% | 0% | 0% | 0% | 0% | 0% | 0% | 0% |
| $f_9$ | ANP | 1(−) | 0.96(−) | 31.40(−) | 39.68(≈) | 28.92(−) | 35.88(≈) | 30.48(−) | 36.44 |
| | PR | 0.004 | 0.004 | 0.14 | 0.18 | 0.13 | 0.16 | 0.14 | 0.16 |
| | SR | 0% | 0% | 0% | 0% | 0% | 0% | 0% | 0% |
| $f_{10}$ | ANP | 1(−) | 1(−) | 9.92(−) | 11.40(≈) | 8.64(−) | 11.44(≈) | 9.68(−) | 11.20 |
| | PR | 0.08 | 0.08 | 0.82 | 0.95 | 0.72 | 0.95 | 0.80 | 0.92 |
| | SR | 0% | 0% | 0% | 56% | 4% | 60% | 0% | 48% |
| −/≈/+ | | 8/2/0 | 8/2/0 | 8/2/0 | 0/10/0 | 7/2/1 | 0/10/0 | 8/2/0 | \ |

**Table 5.** Experimental results of ANP, PR, and Success Rate (SR) obtained by the CMA-ES, MA-ES, CMA-N, CMA-NMO, CMA-NMM, CMA-NMM-MO, MA-ESN, and MA-ESN-MO on $\varepsilon = 0.0001$ for functions $f_{11}$–$f_{20}$.

| Fun. | | CMA-ES | MA-ES | CMA-N | CMA-NMO | CMA-NMM | CMA-NMM-MO | MA-ESN | MA-ESN-MO |
|------|-----|--------|-------|-------|---------|---------|-----------|--------|-----------|
| $f_{11}$ | ANP | 1(−) | 1.04(−) | 3.56(−) | 3.92(≈) | 3.72(−) | 3.96(≈) | 3.60(−) | 4 |
| | PR | 0.16 | 0.17 | 0.60 | 0.65 | 0.62 | 0.66 | 0.60 | 0.66 |
| | SR | 0% | 0% | 0% | 0% | 0% | 0% | 0% | 0% |
| $f_{12}$ | ANP | 1(−) | 1(−) | 2.64(−) | 6.00(≈) | 2.80(−) | 5.40(−) | 2.40(−) | 5.96 |
| | PR | 0.12 | 0.12 | 0.33 | 0.75 | 0.35 | 0.67 | 0.31 | 0.75 |
| | SR | 0% | 0% | 0% | 16% | 0% | 0% | 0% | 4% |
| $f_{13}$ | ANP | 1(−) | 1(−) | 3.20(−) | 3.80(≈) | 3.44(−) | 3.96(≈) | 3.20(−) | 3.92 |
| | PR | 0.16 | 0.16 | 0.54 | 0.63 | 0.57 | 0.66 | 0.53 | 0.65 |
| | SR | 0% | 0% | 0% | 0% | 0% | 0% | 0% | 0% |
| $f_{14}$ | ANP | 1(−) | 0.72(−) | 1.76(−) | 3.52(≈) | 1.72(−) | 3.64(≈) | 1.84(−) | 3.48 |
| | PR | 0.16 | 0.12 | 0.29 | 0.58 | 0.28 | 0.60 | 0.30 | 0.58 |
| | SR | 0% | 0% | 0% | 0% | 0% | 0% | 0% | 0% |
| $f_{15}$ | ANP | 1(−) | 0.76(−) | 1.20(−) | 2.04(≈) | 1.12(−) | 2.40(≈) | 1.32(−) | 2.12 |
| | PR | 0.12 | 0.09 | 0.15 | 0.25 | 0.14 | 0.30 | 0.16 | 0.26 |
| | SR | 0% | 0% | 0% | 0% | 0% | 0% | 0% | 0% |
| $f_{16}$ | ANP | 1(−) | 0(−) | 1.24(−) | 1.96(≈) | 1.16(−) | 2.24(≈) | 1.08(−) | 1.84 |
| | PR | 0.16 | 0 | 0.20 | 0.33 | 0.19 | 0.37 | 0.18 | 0.32 |
| | SR | 0% | 0% | 0% | 0% | 0% | 0% | 0% | 0% |
| $f_{17}$ | ANP | 0.84(−) | 0(−) | 1(−) | 1.48(≈) | 1(−) | 1.44(≈) | 1.04(−) | 1.48 |
| | PR | 0.10 | 0 | 0.12 | 0.18 | 0.12 | 0.18 | 0.13 | 0.19 |
| | SR | 0% | 0% | 0% | 0% | 0% | 0% | 0% | 0% |
| $f_{18}$ | ANP | 0.16(−) | 0(−) | 1(≈) | 1.44(+) | 1(≈) | 1.32(+) | 1(≈) | 1.08 |
| | PR | 0.02 | 0 | 0.16 | 0.24 | 0.16 | 0.22 | 0.16 | 0.18 |
| | SR | 0% | 0% | 0% | 0% | 0% | 0% | 0% | 0% |
| $f_{19}$ | ANP | 0.04(−) | 0(−) | 1(≈) | 1.04(≈) | 1(≈) | 0.80(≈) | 1(≈) | 0.96 |
| | PR | 0.005 | 0 | 0.12 | 0.13 | 0.12 | 0.10 | 0.12 | 0.12 |
| | SR | 0% | 0% | 0% | 0% | 0% | 0% | 0% | 0% |
| $f_{20}$ | ANP | 0(≈) | 0(≈) | 0.04(≈) | 0(≈) | 0(≈) | 0(≈) | 0(≈) | 0 |
| | PR | 0 | 0 | 0.005 | 0 | 0 | 0 | 0 | 0 |
| | SR | 0% | 0% | 0% | 0% | 0% | 0% | 0% | 0% |
| −/≈/+ | | 9/1/0 | 9/1/0 | 7/3/0 | 0/9/1 | 7/3/0 | 1/8/1 | 7/3/0 | \ |

In order to test the statistical significance of the eight compared algorithms, the Wilcoxon's test at the 5% significance level, which is implemented by using KEEL software [42], is employed based on the *PR* values. Table 6 summarizes the statistical test results. It can be seen from Table 6 that the MA-ESN-MO provides higher R+ values than R− values compared with the CMA-ES, MA-ES, CMA-N, CMA-NMM, and MA-ESN. Furthermore, the *p* values of the CMA-ES, MA-ES, CMA-N, CMA-NMO, and MA-ESN are less than 0.05, which means that the MA-ESN-MO is significantly better than these competitors. The *p* values of the CMA-NMO and CMA-NMM-MO are equal to one, which means that the performance of the MA-ESN-MO is not different from that of the CMA-NMO and CMA-NMM-MO. To further determine the ranking of the eight compared algorithms, the Friedman's test, which is also implemented by using KEEL software, is conducted. As shown in Table 7, the overall

ranking sequences for the test problems are the CMA-NMO, CMA-NMM-MO, MA-ESN-MO, CMA-N, CMA-NMM, MA-ESN, CMA-ES, and MA-ES. The experimental results show that the improved algorithms CMA-NMO, CMA-NMM-MO, and MA-ESN-MO perform better than other original algorithms. Therefore, it can be concluded that the improvement strategies are effective. Figure 8 shows the results of ANP obtained in 25 independent runs by each algorithm for functions $f_1$–$f_{20}$ on $\varepsilon = 0.0001$. In order to show these clearly in Figure 8, the CMA-ES, MA-ES, CMA-N, CMA-NMO, CMA-NMM, CMA-NMM-MO, MA-ESN and MA-ESN-MO are abbreviated to CMA, MA, CN, CO, CMM, CMO, MAN, and MMO, respectively.

**Table 6.** Results obtained by the Wilcoxon test for algorithm MA-ESN-MO.

| VS | $R^+$ | $R^-$ | Exact $p$-Value | Asymptotic $p$-Value |
|---|---|---|---|---|
| CMA-ES | 188.5 | 1.5 | $\geq 0.2$ | 0.000144 |
| MA-ES | 208.5 | 1.5 | $\geq 0.2$ | 0.000103 |
| CMA-N | 182.0 | 8.0 | $\geq 0.2$ | 0.00043 |
| CMA-NMO | 52.5 | 137.5 | $\geq 0.2$ | 1 |
| CMA-NMM | 176.5 | 13.5 | $\geq 0.2$ | 0.000911 |
| CMA-NMM-MO | 85.0 | 105.0 | $\geq 0.2$ | 1 |
| MA-ESN | 205.5 | 4.5 | $\geq 0.2$ | 0.000152 |

**Table 7.** Average ranking of the algorithms (Friedman).

| Algorithm | Ranking |
|---|---|
| CMA-ES | 6.8 |
| MA-ES | 7.4 |
| CMA-N | 4.525 |
| CMA-NMO | 2.25 |
| CMA-NMM | 4.75 |
| CMA-NMM-MO | 2.55 |
| MA-ESN | 5 |
| MA-ESN-MO | 2.725 |

Table 8 shows the success rate of all the algorithms in finding all the global optimal solutions from $f_1$ to $f_{10}$. It can be observed that the CMA-NMO, CMA-NMM-MO, and MA-ESN-MO are able to achieve the success rate of 100% on $f_1$–$f_5$. Moreover, the CMA-NMO generates a higher success rate than the CMA-N from $f_6$–$f_{10}$. Similarly, the CMA-NMM-MO and MA-ESN-MO generate higher success rates than the CMA-NMM and MA-ESN from $f_6$–$f_{10}$, respectively. The CMA-ES and MA-ES obtain the success rate of 100% on $f_1$ and $f_3$. However, the CMA-ES and MA-ES generate relatively low success rates on other eight test functions.

**Table 8.** Experimental results of the success rate obtained by the CMA-ES, MA-ES, CMA-N, CMA-NMO, CMA-NMM, CMA-NMM-MO, MA-ESN, and MA-ESN-MO on $\varepsilon = 0.001$

| Fun. | CMA-ES | MA-ES | CMA-N | CMA-NMO | CMA-NMM | CMA-NMM-MO | MA-ESN | MA-ESN-MO |
|---|---|---|---|---|---|---|---|---|
| $f_1$ | 100% | 100% | 100% | 100% | 100% | 100% | 100% | 100% |
| $f_2$ | 20% | 20% | 90.4% | 100% | 100% | 100% | 86.4% | 100% |
| $f_3$ | 100% | 100% | 100% | 100% | 100% | 100% | 96% | 100% |
| $f_4$ | 25% | 25% | 60% | 100% | 67% | 100% | 65% | 100% |
| $f_5$ | 62% | 50% | 80% | 100% | 82% | 100% | 84% | 100% |
| $f_6$ | 5.5% | 5.3% | 71.7% | 98% | 76% | 97.1% | 77.5% | 97.1% |
| $f_7$ | 2.7% | 2.7% | 79% | 83.3% | 77.8% | 80.2% | 81.5% | 83% |
| $f_8$ | 1.3% | 0.7% | 52.8% | 64.1% | 54.6% | 66.1% | 51.1% | 60.9% |
| $f_9$ | 0.4% | 0.4% | 14.4% | 22.2% | 13.7% | 17.6% | 14.6% | 21.1% |
| $f_{10}$ | 8.3% | 8.3% | 85.3% | 97% | 75.6% | 98% | 83.6% | 94.6% |

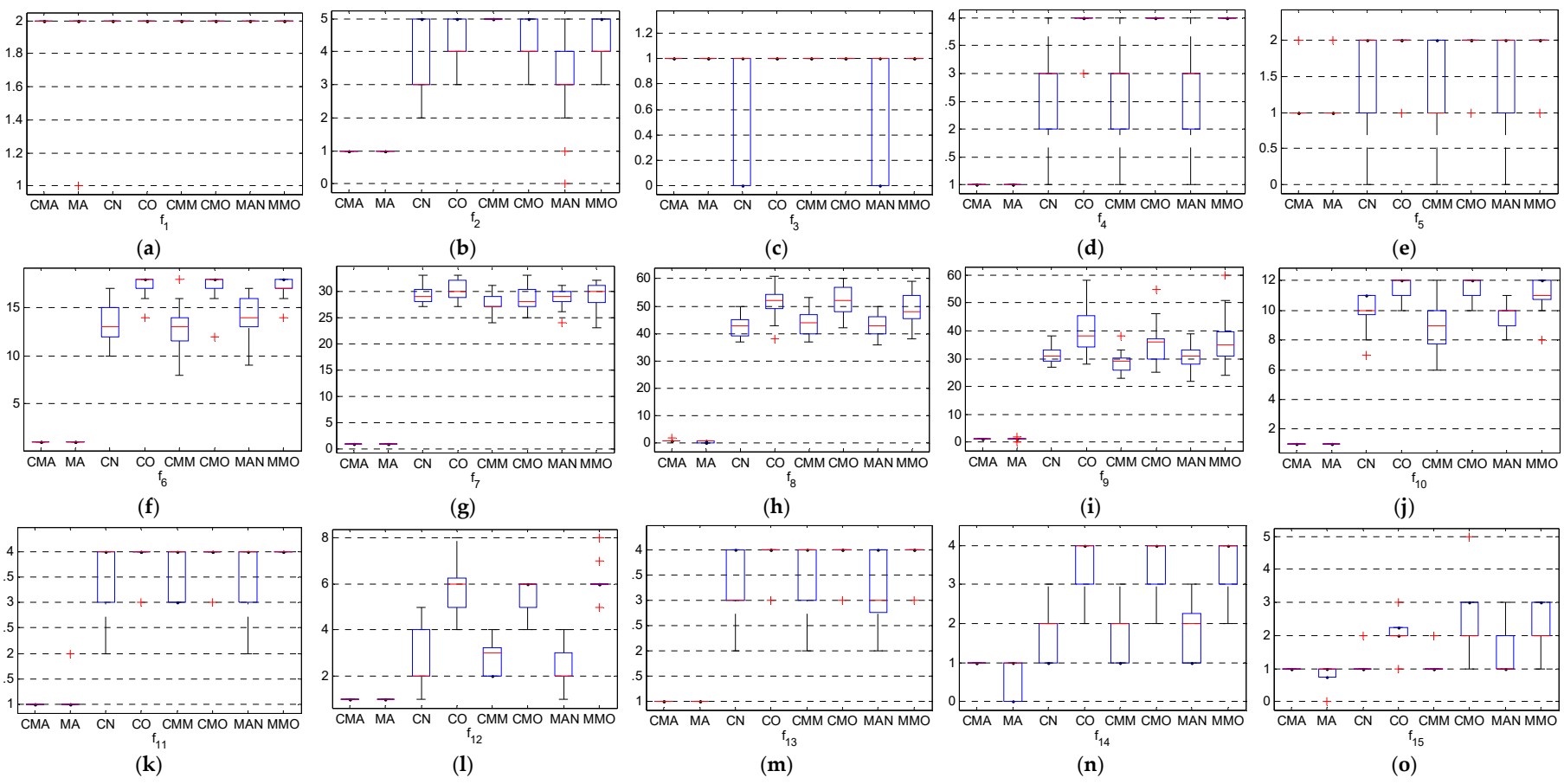

**Figure 8.** *Cont.*

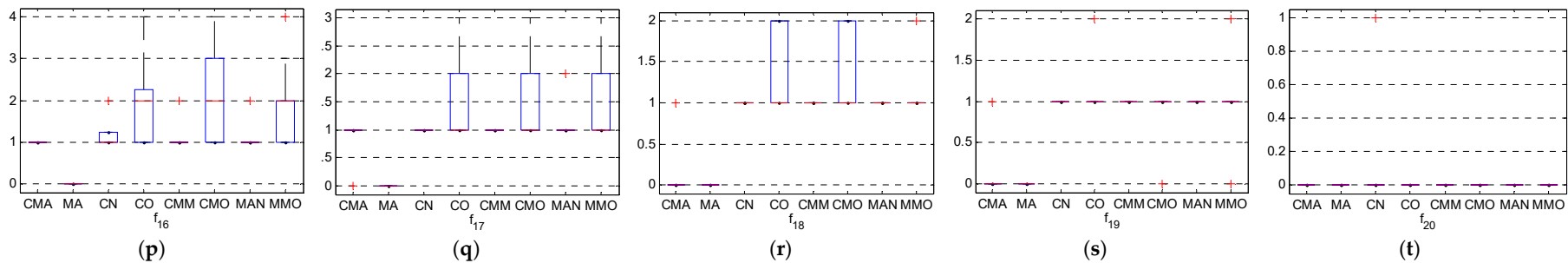

**Figure 8.** Box-plot of peaks found by the CMA-ES, MA-ES, CMA-N, CMA-NMO, CMA-NMM, CMA-NMM-MO, MA-ESN, and MA-ESN-MO on 20 test problems.

Figure 9 displays the average number of peaks in terms of the mean value achieved by each of eight algorithms on $\varepsilon = 0.0001$ for CEC2013 multimodal problems versus the number of *FES*. It can be seen that the CMA-NMO, CMA-NMM-MO, and MA-ESN-MO perform better than other algorithms, which suggests that the mechanism of multi-objective optimization can help the algorithm find multiple global optimal solutions. In addition, the curves of ANP achieved by CMA-ES, MA-ES, CMA-N, CMA-NMM, and MA-ESN are ups and downs instead of steady growth. The reason is that the better parents are discarded after producing the offspring. However, sometimes, the performance of the parent is better than that of the offspring. The curves of the CMA-NMO, CMA-NMM-MO, and MA-ESN-MO show a gradual upward trend. The reason is that the archive is introduced in these algorithms, which is helpful to ensure the convergence of the algorithm.

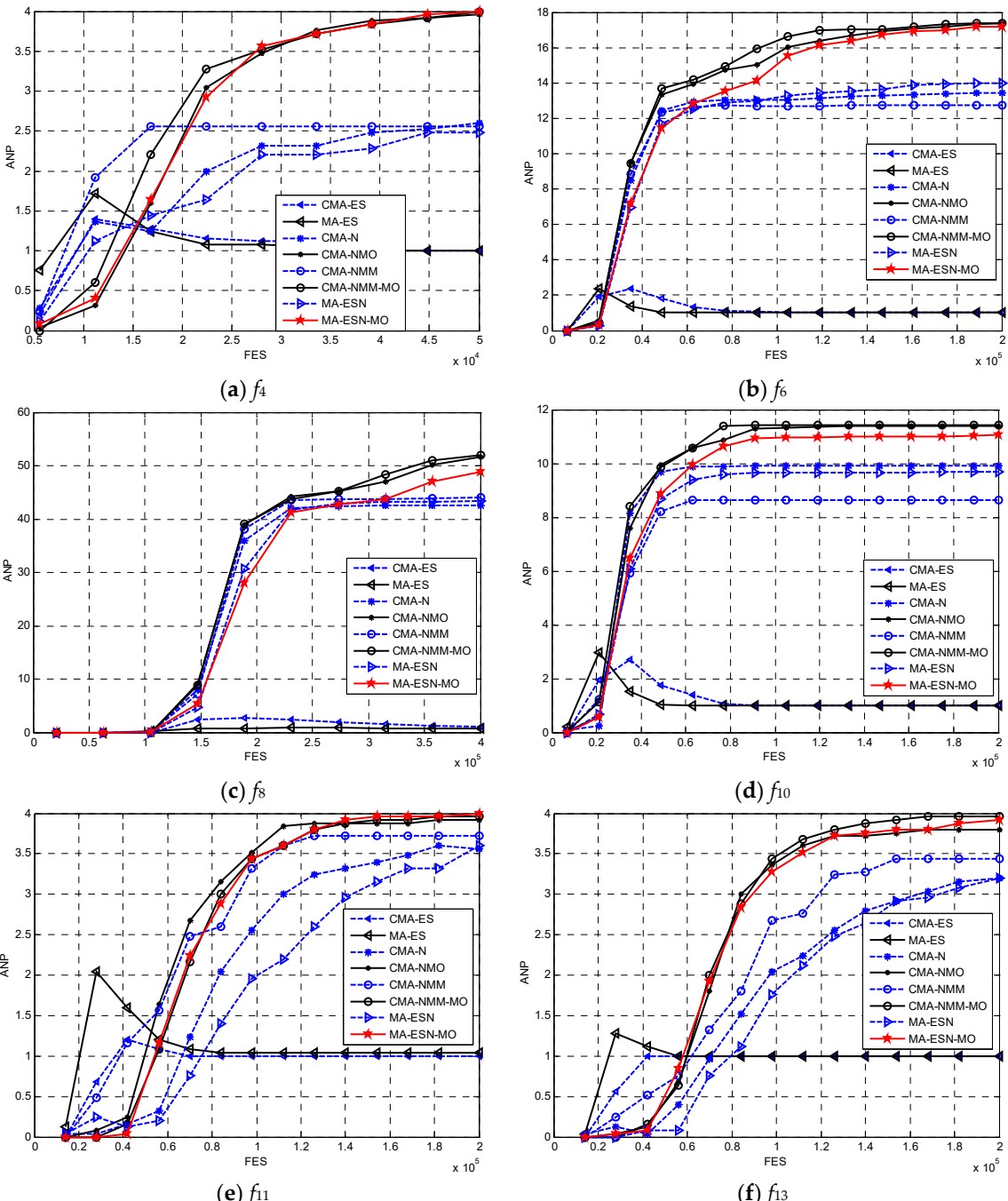

**Figure 9.** *Cont.*

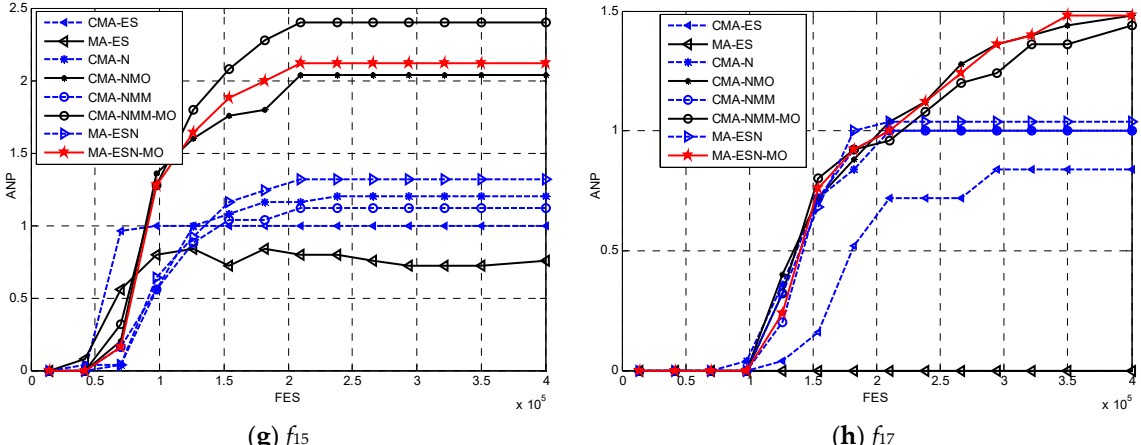

**Figure 9.** Average number of peaks found by the CMA-ES, MA-ES, CMA-N, CMA-NMO, CMA-NMM, CMA-NMM-MO, MA-ESN, and MA-ESN-MO versus the number of *FES* on eight test problems.

## 6. Conclusions

The CMA-ES has received considerable attention as an efficient optimization algorithm. The MA-ES might be more attractive because of a simpler operator compared to the CMA-ES. Although niching techniques have been introduced into the CMA-ES, the performance of the CMA-ES is unsatisfactory for solving multimodal optimization problems. This paper proposed a matrix adaptation evolution strategy with the multi-objective optimization algorithm to solve multimodal optimization problems. The strategy of the multi-objective optimization is used to ensure the diversity of population. The archive is employed to maintain the peak found by the algorithm until the end of the run. The population is divided into multiple subpopulations, which are used to explore and exploit in parallel to find multiple optimal solutions. The experimental results suggest that the proposed strategies can achieve a better performance than the original algorithm on CEC2013 test problems.

**Funding:** This research is partly supported by the Doctoral Foundation of Xi'an University of Technology (112-451116017).

**Acknowledgments:** Thanks to Ofer M. Shir for providing the source code of CMA-N and CMA-NMM.

**Conflicts of Interest:** The author declares no conflict of interest.

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
