# Peer review of "Matrix Adaptation Evolution Strategy with Multi-Objective Optimization for Multimodal Optimization"

_algorithms, doi:10.3390/a12030056_

Round 1
Reviewer 1 Report
After we have the 2nd review, the quality of this paper is enhanced. However, there is one place should be revised. It is quite neccessary to provide the statistic test in the expeirmental result. We saw the Table 4 shows the Wilcoxon signed-rank test according to the peak ratio. However, the statistic test of success rate and ANP are not available. Please provide the test for them.
Finally, a problem of the Wilcoxon signed-rank test doesn't reveal the difference between the two algorithms which are quite closed. For example, the average rank in Table of CMA-NMO and MA-ESN-MO are 1.05 and 1.15, respectively. We are not sure whether there is any difference between them. Hence, we may verify whether they are equal or CMA-NMO is better than the proposed algorithm.
Author Response
We sincerely appreciate the Editor-in-Chief, Associate Editor and the reviewers' constructive comments and feel encouraged by their positive feedback. The concerns of the Editor-in-Chief, Associate Editor and the reviewers for improvement of the manuscript have been carefully addressed. Please find our point-by-point response manuscript entitled "Detailed Response to Reviewers_1".The sentences in italic are the comments, and others are our responses.

Reviewer 2 Report
I have carefully read the resubmitted version and I cannot see any substantial difference with the previous revised version. Thus, I still have the same concerns. The manuscript has still typos and grammar errors (like in line 124), and the core of my previous comments was not adequately addressed, e.g., the comparison with other similar approaches, etc.
Other comments:
line 124: Among some representative work ->Among some representative works
Consider highlighting of particular rows of the table with the simulation results that the author thinks are interesting and the most indicative ones.
Fig 2 should be placed on one page.
line 375 Fig. 8
Author Response
We sincerely appreciate the Editor-in-Chief, Associate Editor and the reviewers' constructive comments and feel encouraged by their positive feedback. The concerns of the Editor-in-Chief, Associate Editor and the reviewers for improvement of the manuscript have been carefully addressed. Please find our point-by-point response manuscript entitled "Detailed Response to Reviewers_2".The sentences in italic are the comments, and others are our responses.

This manuscript is a resubmission of an earlier submission. The following is a list of the peer review reports and author responses from that submission.
Round 1
Reviewer 1 Report
The paper entitled "Matrix Adaptation Evolution Strategy with Multiobjective Optimization for Multimodal Optimization" proposes a new algorithm called matrix adaptation evolution strategy with multiobjective optimization algorithm in order to tackle multimodal optimization problems, accompanied with experimental simulations.
Although the paper exhibits some interesting ideas and I believe that it is worthy, I have several concerns regarding its current form. Below I cite some of these notes.
An important factor that affects the paper's quality is the inadequate language. There are a lot of spelling and grammar mistakes. even the abstract. This is not acceptable for a journal publication. Also, math formulas and typesetting have to be improved, as well.
The discussion of related works must be improved and expanded. Moreover, the technical description of related algorithms should be moved out of this section 2, they might be included in a separate section dedicated to this objective.
The proposed algorithm has to be further explained justifying the differences with the other approaches mentioned in the paper.
Clarification and better presentation for the simulation part are also essential. The tables and their captions are not quite helpful for the reader to grasp their importance.
Again, I believe that the paper's motivation is solid and interesting concepts were included in the paper, therefore I encourage the author to consider my comments and suggestions for a future resubmission.
Author Response
Please find our revised manuscript entitled "Matrix Adaptation Evolution Strategy with Multiobjective Optimization for Multimodal Optimization_ 416905_revised ".
We sincerely appreciate the Editor-in-Chief, Associate Editor and the reviewers' constructive comments and feel encouraged by their positive feedback. The concerns of the Editor-in-Chief, Associate Editor and the reviewers for improvement of the manuscript have been carefully addressed. Below we provide our responses to the comments. The sentences in italic are the comments, and others are our responses.
If you have any questions, you can contact us via email (Mrs. Wei Li, liwei@ xaut.edu.cn or [email protected]).
Thanks very much for your attention to our paper. We are looking forward to your response.
Kind regards,
Wei Li
School of Computer Science and Engineering, Xi’an University of Technology

Reviewer 2 Report
This paper is well-written and the transformation of the multimodal optimization problem is promising. However, there are some comments given by this reviewer. Please refer them below.
1. This paper studied the multimodal optimization problem which was transformed into a bi-objective problem. It is an interesting aspect to know this transformation.We saw this paper cited the paper from 15 to 19. This reviewer may suggest we add the brief descriptions of these works. In addition, the following paper was quite related to this work. This paper could be included in the reference.
Cheng, R., Li, M., Li, K., & Yao, X. (2018). Evolutionary Multiobjective Optimization-Based Multimodal Optimization: Fitness Landscape Approximation and Peak Detection. IEEE Transactions on Evolutionary Computation, 22(5), 692-706.
2. Even though the authors advocated that it is a difficult work to transform the multimodal problem into the MO problem, please state the differences between this work with these references.
3. Because there are 20 original multimodal problems from the CEC 2013, please draw some functions before we show the experimental results.
4. The Algorithm 1 and Algorithm 2 have less descriptions. Please describe these pseduo code in the Section 3.2.
5. There are no equation number in Section 4.1.
6. Even though the multimodal problems are transformed into a multi-objective problem, we can't see their Pareto fronts in the diagram.
Author Response

(The authors gave the same response as above.)

Round 2
Reviewer 1 Report
To begin with, I thank the author for taking into consideration parts of my previous comments. The revised manuscript has some improvements over the initial submission, several mistakes are now corrected, and the overall paper's structure is better.
But the author has not responded to all my previous concerns. Although many typos and grammar errors are now corrected, there are still plenty of them present throughout the text.
Differences with other similar approaches are still absent and the description of the algorithm has not substantially changed. The same holds for the simulation part. Vital parameters and variables that could potentially affect the results are not stated.
Overall, the author has improved the paper but only partially. It still needs heavy revision, therefore I recommend rejection, Nevertheless, I encourage a future resubmission, there are some interesting ideas that can be better presented.
Reviewer 2 Report
This reviewer may sincerely suggest the author should illustrate the revisions directly in the response to reviewers. In particularly, this reviewer likes to differentitate the difference between the proposed algorithms to the existing papers. So it may save the time for reviewers to double check the content. Please simply enhance the response to the reviewer this time without modifying the manuscript. Thank you.